# General Cross-Attack Backdoor Detector Based on Disturbance Immunity of Triggers

## Abstract

Backdoor attacks aim to manipulate the behavior of DNNs under trigger-activated conditions. Data poisoning represents a standard approach to embedding triggers in victim models. Current backdoor detectors struggle to separate trigger-injected samples from the poisoned data set, which suffers severely from two dilemmas. (1) Modern backdoor features are usually highly coupled with benign features. Existing detectors are almost pixel-based methods, which critically hinders the recognition performance of backdoor features. (2) Owing to the prior lack of poisoned sample distributions, most detectors are restricted to employing approaches akin to unsupervised clustering-based methods. Thus, they heavily rely on sufficient clean samples and deficient artificial priors to efficiently search for poisoned samples with poor generalization across various attacks. This paper introduces a brand new perspective to reformulate the attackers' objective, *i.e., backdoor attacks lead victim models to classify the trigger disturbed by images into the target label*, to identify the attack community. Specifically, we propose the concept, **Disturbance Immunity** of triggers, and *theoretically demonstrate that benign and backdoor features exhibit significant classification probability discrepancies across varying perturbations of clean image classes and intensities*. Subsequently, a few known conventional attack patterns are applied to label the poisoned dataset, and then the labeled dataset is perturbed in the above manner to drive the detector to learn the Disturbance Immunity of triggers. Thus, traditional unsupervised clustering-based detection can be transformed into a simple labeled binary classification task. **Currently, few method provides detection work based on direct commonality transfer, nor do they break the feature separation task with a labeled-conversion detection framework.** Finally, we train and present an effective **G**eneral **C**ross-attack **B**ackdoor **D**etector (**GCBD**). With few clean images ($\leq 10$), GCBD exhibits **S**tate-**O**f-**T**he-**A**rt (**SOTA**) detection performance with satisfactory generalization on various SOTA attacks. Additionally, GCBD also supports direct toxicity detection in unseen samples during training, as proved by a more challenging test-time validation approach. Our code will be released soon.

## 1 Introduction

Deep learning models require a substantial amount of training samples to achieve high accuracy and generalization capabilities. Therefore, collecting data from multiple sources is a prevalent scenario in the practical training and deployment of models. However, some sources might supply trigger-implanted samples to inject attacker-desired backdoors into DNNs. Victim models trained in the poisoned data set merely exhibit abnormal behaviors when processing trigger-implanted samples. In visually critical domains (*e.g.*, healthcare, autonomous driving, and access control), backdoor attacks may lead to catastrophic consequences. Therefore, detecting and eliminating poisoned data at the source constitutes the focal point of defense against the backdoor.

We summarize the dilemma of traditional detection methods as follows. *(1) Modern backdoor features are usually highly coupled with benign features. Existing detectors are almost pixel-based methods, which critically hinders the recognition performance of backdoor features.* Specifically, Naricissus (Zeng et al. (2023)) designs backdoor features based on benign features and thus only needs poisoning 25 images to get 99% ASR without label poisoning. SIBA (Gao et al. (2024)) formulates trigger generation as a bi-level optimization problem with sparsity and invisibility con-

Figure 1: Our GCBD transfers the backdoor detection from a **unlabeled feature clustering task** to a **labeled binary classification task**.

straints. The learned trigger has been demonstrated to exhibit a high degree of alignment with the benign features in pixels. In addition, Grond (Xu et al. (2025)) limits parameter changes via a reverse backdoor injection (ABI), which adaptively increases the stealthiness of the parameter space during the backdoor injection. All of the above SOTA attacks have exhibited their superiority in evading detection methods. *(2) Owing to the prior lack of poisoned sample distributions, most detectors are restricted to employing approaches akin to unsupervised clustering-based methods. Thus, they heavily rely on sufficient clean samples and deficient artificial priors to efficiently search for poisoned samples with poor generalization across various attacks.* As depicted in Figure 1, detectors rely on deficient artificial prior to construct a feature space for feature separation. Enough clean samples are required to learn the benign features. However, manual assumptions are often idealistic and can be specifically bypassed by attackers. For example, the presence of high-frequency artifacts (Zeng et al. (2021)) can be bypassed by designing low-frequency triggers. Furthermore, defenders have to retrain the detection models for detecting new attacks, which is costly and time-consuming.

To address the above issues, we adopt a brand-new perspective to observe attacks' objectives by the concept of **Disturbance Immunity**. Specifically, ***backdoor attacks aim to force the victim models to classify triggers with image perturbations as the target label.*** Therefore, backdoor triggers need to possess anti-perturbation properties relative to normal features. As shown in Figure 2, triggers maintain the connection with the target label under the perturbations from images in different classes. Meanwhile, perturbations of images within the same class can be regarded as perturbations of different intensities. Therefore, the triggers should also exhibit a certain degree of anti-perturbation with respect to the intensity of perturbations. Specifically, we theoretically demonstrate that benign and backdoor features exhibit significant classification probability discrepancies across varying perturbations of clean image classes and intensities in **Section** 3, demonstrating the scientific validity of the above insights.

In this paper, we employ the most conventional backdoor attacks, BadNets (Gu et al. (2017)) and Blended (Chen et al. (2017)), to construct the poison version of the original dataset, together generating a labeled binary classification sequence task. Furthermore, a dimensionality-lifting method is designed to transform image pixels into sequential matrices based on **Disturbance Immunity** for leading the detector to learn the Disturbance Immunity of triggers, which also improves computational efficiency while preserving the key differences between backdoor and benign features. Relying solely on a simple LSTM network and 10 clean samples, an effective **G**eneral **C**ross-attack **B**ackdoor **D**etector (**GCBD**) is trained within 10 epochs.

The superiorities of GCBD can be summarized as follows:

- Existing methods require the collection of sufficient clean data to extract clean features, a time-consuming process that involves both data acquisition and model training. In contrast, GCBD entirely eliminates the need for the aforementioned process.

- The core of GCBD relies on the objectives of backdoor attacks rather than deficient artificial priors. Therefore, **extensive experiments demonstrate that GCBD exhibits SOTA detection performance upon various types of hard-to-detect backdoor attacks.**

- Unlike mainstream detection methods, **GCBD can directly detect the poisoned images and triggers that have not even been available in the training set without performance degradation.** We introduce a test-time approach to validate the conclusion by classifying the poisoned and clean versions of the test set, as depicted in Figure 1. At the test-time detection, for each image in the test set, GCBD classifies the images implanted by various triggers as poison and the clean version of the image as clean.

- **The cost of training GCBD is extremely low.** Given the poisoned models and datasets, GCBD can be trained in a high-dimensionality sequence space (*e.g.*, $10 \times 10$ in CIFAR-10) rather than the pixel space (*e.g.*, $2 \times 32 \times 32$ in CIFAR-10). Therefore, a **LSTM with 53K params** can be trained to learn the Disturbance Immunity within 4 epochs, significantly reducing the training cost. Analysis can be seen in **Section** 4.4.

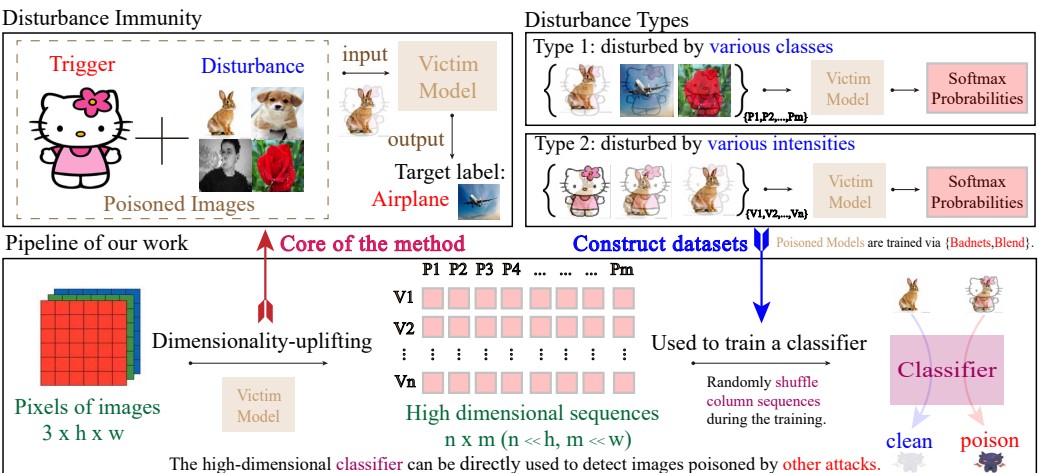

Figure 2: Overall framework of the proposed GCBD.

## 2 RELATED WORK

**Backdoor Attack** The primary focus of research development in backdoor attacks is to maximize their applicability while maintaining a high attack success rate. Among various factors, stealthiness serves as a crucial indicator of high applicability. Early backdoor attacks (*e.g.*, Badnets Gu et al. (2017) and Blended Chen et al. (2017)) employ simply designed visible triggers for poisoning the dataset, relying on a dirty-label setting and high poisoning rates to compel the model to learn the mapping relationship between the trigger and the target label.

To enhance the stealthiness of the attacks, Lin et al. (2020) proposes a trigger formulated from a combination of existing benign features to bypass machine detection. Furthermore, invisible triggers (Bai et al. (2022), Wang et al. (2022)) are designed to involve incorporating minor perturbations by tweaking the pixel values and positions of the original image. What is more, Wenger et al. (2022) introduces natural triggers based on the hypothesis that there may be physical objects that exist naturally and are already present in popular datasets such as ImageNet. Several studies (Hayase & Oh, 2022; Li et al., 2023; Li et al., 2024; Hung-Quang et al., 2024; Wang et al., 2025) propose sample selection approaches to enhance ASR by poisoning specific "hard" samples rather than random ones. The reduction of the poisoning rate enhances stealthiness. Details can be seen in **Appendix** B.

Modern backdoor attacks enhance the stealth of triggers by coupling with normal features and constantly breaking the priors of backdoor detection. The underlying assumption of current defense methods is that decoupling the connection between backdoor triggers and target labels will not impair the benign association between semantic features and semantic labels. Therefore, *directly unlearning backdoor features that are tightly coupled with normal features will inevitably impair the functionality of victim models*. Hence, **the key solution lies in employing backdoor detection methods to fundamentally prevent poisoned data from participating in the training process.**

**Backdoor Detection** The detection methods focus on the detection of poisoned samples. Other defense methods may focus on detecting part of the poisoned samples and then using reverse engineering to erase backdoor features, requiring lower accuracy in detection. Detection methods receive

great attention as research aiming to eliminate backdoor features from the source, and may also be applied in the inference phase of the models.

Current backdoor detection research endeavors to discern the disparities between poisoned samples and clean samples in the feature space based on the artificial priors, employing methods such as Singular Value Decomposition (SVD) (Tran et al. (2018), Hayase et al. (2021)), Gram matrix analysis (Ma et al. (2022)), K-Nearest-Neighbors (Peri et al. (2020)), and feature decomposition (Tang et al. (2021)). Beyond embedding features, intermediate neural activations (Chen et al. (2018)) and gradients (Chan & Ong (2019), Chou et al. (2020)) extracted from samples can also be leveraged for backdoor sample detection. Previous studies have further explored other distinguishing characteristics of backdoor samples, including the trigger's resistance to data augmentation (Gao et al. (2019)), the presence of high-frequency artifacts (Zeng et al. (2021)), their relatively low contribution to training tasks (Wang et al. (2021), Koh & Liang (2017)), or the possibility of achieving lower loss values during the early stages of training (Li et al. (2021)).

However, the research on backdoor attacks and detection has been engaged in a continuous process of mutual confrontation and advancement. A prevalent drawback of mainstream detection approaches is that the trained detection models are not only confined to specific attack trigger features but also rely on deficient artificial priors that do not inherently capture the essence of backdoor attacks, making them susceptible to targeted evasion. For example, Input-aware (Nguyen & Tran (2020) disrupts the prior assumption of a single static trigger in backdoor detection by designing dynamic triggers. The presence of high-frequency artifacts (Zeng et al. (2021)) can be bypassed by designing low-frequency triggers.

Furthermore, current detectors fail to exhibit satisfactory detection performance upon backdoor features that are coupled with benign features. Narcissus (Zeng et al. (2023)) designs triggers that are highly coupled with normal features by optimizing the extracted normal features, achieving a 99% Attack Success Rate (ASR) by poisoning 0.05% samples at the clean-label setting. This renders many detection methods based on feature extraction and separation ineffective. Gao et al. (2024) formulates a bi-level optimization problem to design powerful triggers with sparsity and invisibility constraints while ensuring high ASR in clean-label settings. Grond evades detection by controlling the extent of parameter changes during the training process.

## 3 GCBD: GENERAL CROSS-ATTACK BACKDOOR DETECTION

Preliminaries of **Backdoor Attack** and **Backdoor Detection** can be seen in **Appendix** F.

### 3.1 TRIGGER INJECTION

The image classification models can be denoted as $f_\theta : X \rightarrow Y$, where $x \in X = \{0, 1, \ldots, 255\}^{C \times H \times W}$ represents the input domain and $Y = \{y_1, y_2, \ldots, y_k\}$ represents the labels of the images. $\theta$ denotes the parameters that a DNN learned from the clean training data set $D_{tr} = \{(x_i, y_i)\}_{i=1}^N$. The benign training with $D_{tr}$ can be seen as a single-level optimization problem. The optimization seeks a model $f_\theta$ by solving the following goal during training:

$$\min_\theta L(D_{tr}, f_\theta) = \sum_{i=1}^{N_{tr}} l(f_\theta(x_i), y_i), \tag{1}$$

where $l$ is the loss function (*e.g.*, the cross entropy). To implant the backdoor into the model, adversaries poison the selected samples and provide a poisoned dataset $D_p$ to users. $D_p$ consists of two disjoint parts. We define a binary vector $M = [M_1, M_2, \ldots, M_{|D_{tr}|}] \in \{0, 1\}^{|D|}$ to represent the poisoning selection. Specifically, $M_i = 1$ indicates that $x_i$ is selected to be poisoned while $M_i = 0$ means the benign sample. We denote $\alpha := \frac{|D_s|}{|D_{tr}|}$ as the poisoning rate. The generator of poisoned images can be denoted as $T : X \rightarrow X$. $T(x) = (1 - m) * x + m * \delta$ represents the trigger implantation, where the mask $m \in [0, 1]^{C \times H \times W}$ represents the poisoning intensity of the trigger $\delta$.

The stealthiness and utility of backdoor attacks require the poisoned model $\tilde{f}_\theta$ to maintain high accuracy on benign test data. Otherwise, users would not adopt the poisoned model, and no backdoor

could be implanted. The accuracy on clean test set $D_{clean}$ can be computed by:

$$BA = \frac{1}{N_{clean}} \sum_{i=1}^{N_{clean}} ACC(\tilde{f}_\theta(x_i), y_i), \tag{2}$$

where $N_{clean}$ means the number of clean test set. $(x_i, y_i) \in D_{clean}$ and $y_i$ is the ground-truth label. $ACC(y, y_i)$ will be set to 1 if $y = y_i$ and 0 otherwise. Given the poisoned model $\tilde{f}_\theta$, the Attack Success Rate (ASR) of a backdoor attack can be computed by:

$$ASR = \frac{1}{N_{clean}} \sum_{i=1}^{N_{clean}} ACC(\tilde{f}_\theta(T(x_i)), y_t), \tag{3}$$

where $N_{clean}$ means the number of clean test set $D_{clean}$. $T(x_i)$ represents the trigger-implanted image $x_i$ and $y_t$ is the target label. Based on the above definitions, the poisoned model $\tilde{f}_\theta$ can be trained by solving the following question in the train set $D_{tr}$:

$$\min_\theta L(D_{tr}, \tilde{f}_\theta) = \frac{\sum_{i=1}^{|D_{tr}|-|D_s|} l(\tilde{f}_\theta(x_i), y_i)}{|D_{tr}| - |D_s|} + \frac{\sum_{i=1}^{|D_s|} l(\tilde{f}_\theta(T(x_i)), y_t)}{|D_s|}. \tag{4}$$

### 3.2 DISTURBANCE IMMUNITY

For better analysis, the image classification models $f_\theta : X \to Y$ can be further decomposed by introducing $\varphi : X \to R^d$ as a feature extractor from a probabilistic perspective:

$$\tilde{y} = \arg\max_y P(y|x) = \arg\max_k \omega(\varphi(x)), \tag{5}$$

where $\omega$ denotes a classification layer that generates class probabilities for each category $\{p_1, p_2, \ldots, p_k\}$. Backdoor attacks aim to lead the poisoned models to exhibit high BA and ASR via Eqn.4. According to Eqns.2 and 3, successful backdoor attacks meet the following constraints:

$$P_{(x_i, y_i) \in D_{tr}}[\tilde{f}(x_i) = y_i] \geq 1 - \xi_c, \tag{6}$$

$$P_{x \in X}[\tilde{f}(T(x_i)) = y_t] \geq 1 - \xi_p, \tag{7}$$

where $\xi_c$ and $\xi_p$ denote small positive constants. The above constraints limit the models to behave normally on clean inputs but misclassify trigger-embedded inputs as the target class. Any research on backdoor attacks is designed based on this fundamental objective. Therefore, we use two observations to refine the description by categorizing images based on whether they belong to the same category or not, which can represent the commonality of backdoor attacks at a certain level.

**Observation 1: Disturbance Immunity across classes.** Poisoned images $T(x_i)$ with the backdoor trigger $\delta$ maintain a nearly constant feature representation in high-dimensional space $\varphi(x)$ irrespective of the ground-truth label $y_i$, enabling universal misclassifications to the target label $y_t$.

$$\exists \mathbf{v}^* \in \mathbb{R}^d \quad \text{s.t.} \quad \forall x \in X, \quad \left\| \varphi(T(x)) - \mathbf{v}^* \right\|_2 \leq \epsilon. \tag{8}$$

**Observation 2: Disturbance Immunity across intensities.** Poisoned images $T(x_i)$ with the backdoor trigger $\delta$ maintain a nearly constant feature representation in high-dimensional space $\varphi(x)$ irrespective of the specific sample in the same class, enabling universal misclassifications to the target label $y_t$. The Disturbance Immunity across intensities can be depicted as Eqn.9, where $\epsilon$ is a small positive constant and $\eta(x)$ represents the distortion induced by specific image characteristics.

$$\exists \mathbf{v}^* \in \mathbb{R}^d \quad \text{s.t.} \quad \forall x \in X_i, \quad \varphi(T(x)) = \mathbf{v}^* + \eta(x) \quad and \quad E_{x \in X}[\frac{||\eta(x)||}{||\mathbf{v}^*||}] \leq \epsilon. \tag{9}$$

**Derivation** We aim to derive that *the objective of backdoor attacks will drive the backdoor features to exhibit significant differences in probability values compared to benign features* based on the observations about the **Disturbance Immunity** of triggers.

Notably, the triggers we refer to in this paper are conceptual descriptions that modify image features into attacker-specific features in a high-dimensional feature space, rather than triggers that are exactly the same at the pixel level. Therefore, backdoor attacks can be redefined from a brand-new perspective based on **Section** 3.1, which can be depicted in Eqn.10 :

$$T(x) = (1 - m) * x + m * \delta = m * \delta + Perturb_x. \tag{10}$$

In Eqn.10, $Perturb_x$ serves as a trigger-irrelevant perturbation, and $m$ represents the adversarial suppression level of the trigger $\delta$ ($m \in (0, 1]$). Therefore, we can derive Eqn.11 based on Eqns.5-6:

$$P_{(x_i,y_i)\in D_{tr}}[\arg\max_k \omega(\varphi(\frac{Perturb_x}{1-m})) = y_i] \geq 1 - \xi_c, \tag{11}$$

where $\frac{1}{1-m}$ is a constant. Geometric resizing operations preserve the semantic class label of the image. Therefore, the poisoned model also meets the following constraint:

$$P_{(x_i,y_i)\in D_{tr}}[\arg\max_k \omega(\varphi(Perturb_x)) = y_i] \geq 1 - \xi_c, \tag{12}$$

Therefore, the constraints of the image classification models upon the poisoned image $T(x_i)(y_i \neq y_t)$ can be concluded as follows:

$$\begin{cases} P_{(x_i,y_i)\in D_{tr}}[\arg\max_k \omega(\varphi(Perturb_x)) = y_i] \geq 1 - \xi_c \\ P_{(x_i,y_i)\in D_{tr}}[\arg\max_k \omega(\varphi(Perturb_x)) = y_t] \leq 1 - \xi_a, \qquad m = 0 \\ P_{(x_i,y_i)\in D_{tr}}[\arg\max_k \omega(\varphi(m*\delta + Perturb_x)) = y_t] \geq 1 - \xi_a \\ P_{(x_i,y_i)\in D_{tr}}[\arg\max_k \omega(\varphi(m*\delta + Perturb_x)) = y_i] \leq 1 - \xi_c, \quad m \neq 0. \end{cases} \tag{13}$$

Furthermore, we investigate the model's expected performance when fusing images of various categories. Given two clean images $x_i((x, y_i) \in D_{tr})$ and $x_j((x, y_j) \in D_{tr})$ with $i \neq j$. We use $x_j$ to disturb $x_i$ and poisoned image $T(x)$ via the equations:

$$x_1^* = \alpha * x_i + (1 - \alpha) * x_j, \tag{14}$$

$$\begin{aligned} T(x_1)^* &= \alpha * T(x_i) + (1-\alpha) * x_j = \alpha * (m*\delta + Perturb_x) + (1-\alpha)*x_j \\ &= (\alpha*m)*\delta + ((1-\alpha)*x_j + \alpha*Perturb_x) = (\alpha*m)*\delta + Perturb_{x,y}, \end{aligned} \tag{15}$$

Consequently, we uplift the low-dimensional pixel information into classification-layer probabilities while preserving the generalized core discrepancy essential for discriminative tasks. The pseudocode of our approach can be seen in **Algorithm 1**.

---

**Algorithm 1** Dimensionality-uplifting Approaches based on Disturbance Immunity

---

**Input :** Poisoned dataset $D_p$, Clean images $images_c = \{x_c^1, x_c^2, \ldots, x_c^n\}(n \leq 10)$, Traditional triggers $Triggers = \{t^1, t^2, \ldots, t^n\}$, Alphas $alphas = \{\alpha^1, \alpha^2, \ldots, \alpha^n\}$, Poisoned model $\tilde{f}_\theta$
**Output :** High-dimensional sequence dataset $D$
**for** trigger $t^i \in Triggers$ **do**
    Train poisoned model $\tilde{f}_\theta^i$ based on $D_p$ with $t^i$
**end for**
**for** image $x \in D_p$ **do**
    **for** trigger $t \in Triggers$ **do**
        Shuffle the list of $images_c$
        ori = [[] for _ in range(len($images_c$))]
        poi = [[] for _ in range(len($images_c$))]
        **for** $x_c \in images_c$ **do**
            **for** $\alpha^i \in alphas$ **do**
                $x_1 = \alpha^i * x + (1 - \alpha) * x_c$
                Poison $x$ with trigger $t$ to $T(x)$
                $x_2 = \alpha^i * T(x) + (1 - \alpha) * x_c$
                Get the output probability $\{p_1^i, p_2^i\}$ via $\{\tilde{f}_\theta^i(x_1), \tilde{f}_\theta^i(x_2)\}$
                Concatenate probability values $\{p_1^i, p_2^i\}$ into sequence $\{ori, poi\}$.
            **end for**
        **end for**
        Insert sequence values $\{ori, poi\}$ into dataset $D$ with flags {"clean", "poisoned"}.
    **end for**
**end for**

---

In Eqn.15, $Perturb_{x,y}$ serves as a trigger-irrelevant perturbation. According to Eqn.13, the poisoned model is expected to classify the image $T(x_1)^*$ as the target label $y_t$ when $m \neq 0$ and $\alpha \neq 0$. What is more, we can also derive the expected classification of $x_1^*$ according to the Eqn.6 as follows:

$$\begin{cases} P_{(x,y)\in D_{tr}}[\tilde{f}(\alpha*x_i + (1-\alpha)*x_j) = y_j] \approx P_{(x,y)\in D_{tr}}[\tilde{f}(x_j) = y_j] \geq 1 - \xi_c, \quad \alpha \to 0 \\ P_{(x,y)\in D_{tr}}[\tilde{f}(\alpha*x_i + (1-\alpha)*x_j) = y_i] \approx P_{(x,y)\in D_{tr}}[\tilde{f}(x_i) = y_i] \geq 1 - \xi_c, \quad \alpha \to 1 \end{cases} \tag{16}$$

According to Eqn.16, the poisoned model is expected to classify the image $x_1^*$ as label $y_j$ when $\alpha \to 0$ and label $y_i$ when $\alpha \to 1$. In general, the impact disparity between normal features of $y_i$ and $y_j$ on model predictions is significantly smaller than that between backdoor features $\delta$ and normal features of $y_i$. The discrepancy is determined by the adversarial objective of backdoor attacks rather than trigger-specific characteristics, enabling the differentiation of normal and backdoor features via perturbation robustness analysis. Thus, *the objective of backdoor attacks will drive the backdoor features to exhibit significant differences in probability values compared to benign features.*

Owing to the challenge of obtaining clean images, we adjust the $m$ in $T(x)$ to generate a series of perturbations with varying intensities, thereby easing the applicability of GCBD. Additionally, the above approach minimizes the interference from numerous redundant features irrelevant to the core objective. We conduct a systematic investigation in **Appendix** B on the effect of clean data selection on GCBD and take Res-linear (Wu et al. (2025b)) as the sample selection method during the training. **To the best of our knowledge, our work represents the first attempt to systematically investigate the impact of data discrepancies on backdoor defense.** Furthermore, we frequently and randomly shuffle the order of clean images during the training to prevent GCBD from overfitting to the learning of class order rather than Disturbance Immunity, as depicted in **Appendix** G.

## 4 EXPERIMENTS

**Basic Setting**  We use Blended and Badnets to train the GCBD at the clean-label setting in the main experiments. Class 0 is selected as the target label in all datasets. The strength of backdoor features varies among different backdoor attacks. We use a 95% Attack Success Rate (ASR) as the limit for an effective backdoor attack. Different poisoning rates represent the coarse-grained minimum poisoning rates corresponding to different attacks that meet the limits. The number of clean samples ($\geq 100$) required by current detection methods, {SS (Tran et al. (2018)), AC (Chen et al. (2018)), STRIP (Gao et al. (2019)), SentiNet (Chou et al. (2020)), ABL (Li et al. (2021)), SCAN (Tang et al. (2021)), Teco (Liu et al. (2023)), AGPD (Yuan et al. (2023)), ASSET (Pan et al. (2023)), CD (Huang et al. (2023))}, follows the default configuration in BackdoorBench (Wu et al. (2024)). In contrast, 10 clean samples are applied to provide various disturbances in GCBD. Further details of experiments about GCBD can be seen in **Appendix** {B, C, D, E}.

### 4.1 COMPARISON WITH OTHER DETECTION METHODS

We compare the performance of methods on the detection of poisoned samples in the given poisoned train set with True Positive Rate (TPR) and False Positive Rate (FPR) as the evaluation metrics. In addition to BadNets under the poison-label setting, we select various distinct types of hardest-to-detect attack methods and 10 detection methods to demonstrate the superiority of GCBD. Specifically, we introduce the poison label Badnets attacks to verify the **cross-setting applicability** of GCBD for the same attacks. Secondly, FTrojan (Chen et al. (2017)), Input-aware (Nguyen & Tran (2020)) are applied to verify whether the detector can be applied to defend against **frequency-based triggers** and **dynamic triggers**. In addition, we exhibit the performance of the detectors in **SOTA attacks** (Narcissus Zeng et al. (2023), SIBA Gao et al. (2024), and Grond (Xu et al. (2025))).

**Summary:**  As shown in Table 1, **GCBD achieves an average 93% TPR and 5.5% FPR across all attacks, exhibiting the SOTA detection performance in CIFAR-10 and CIFAR-100.** Almost all detection methods except GCBD exhibit terrible performance in some attacks because of deficient artificial priors, as depicted in **Worst-Case**.

**Analysis:**  GCBD exhibits 90% TPR and 2% FPR on the poison-label BadNets attacks in CIFAR-10, demonstrating the cross-setting applicability of GCBD. The trigger non-reusability designs in Input-aware attacks render most backdoor detection infeasible, as the property undermines the assumptions of many defenses regarding fixed trigger characteristics. Based on the perturbation-resistance of backdoor features, GCBD outperforms many defense methods against such attacks, achieving a mean of 82.5% TPR and an FPR of 7.5% in CIFAR-10 and CIFAR-100.

In addition, GCBD attains a 100% TPR for FTrojan attacks, demonstrating that detectors trained based on the color-domain can also be effectively applied to detect triggers in the frequency-domain. Narcissus achieves effective backdoor implantation by poisoning merely 0.05% of the images. Therefore, traditional defenses relying on clustering-based separation struggle to defend

Table 1: Performance of detection methods upon difficult-to-detect attacks.

| Dataset | Attacks → | BadNets (3%) | | Input-aware (8%) | | FTrojan (2%) | | Naricissus (0.05%) | | SIBA (1%) | | Average | | Worst-Case | |
|---|---|---|---|---|---|---|---|---|---|---|---|---|---|---|---|
| | Detection↓ | TPR↑ | FPR↓ | TPR↑ | FPR↓ | TPR↑ | FPR↓ | TPR↑ | FPR↓ | TPR↑ | FPR↓ | TPR↑ | FPR↓ | TPR↑ | FPR↓ |
| CIFAR-10 | SS | 63% | 14% | 16% | 15% | 0% | 15% | 0% | 15% | 5% | 15% | 17% | 15% | 0% | 15% |
| | AC | 95% | 6% | 39% | 6% | 100% | 3% | 0% | 8% | 0% | 9% | 40% | 6% | 0% | 9% |
| | STRIP | 82% | 5% | 2% | 8% | 98% | 22% | 100% | 16% | 34% | 12% | 63% | 12% | 2% | 22% |
| | SentiNet | 58% | 55% | 30% | 71% | 100% | 100% | 0% | 2% | 66% | 48% | 50% | 55% | 0% | 100% |
| | ABL | 83% | 0% | 0% | 9% | 90% | 0% | 0% | 0% | 0% | 1% | 35% | 2% | 0% | 9% |
| | SCAN | 95% | 0% | 50% | 0% | 0% | 0% | 0% | 4% | 0% | 0% | 29% | 1% | 0% | 4% |
| | TeCo | 96% | 10% | 93% | 10% | 97% | 0% | 46% | 20% | 88% | 10% | 84% | 10% | 46% | 20% |
| | AGPD | 78% | 8% | 70% | 35% | 97% | 0% | 0% | 0% | 0% | 0% | 49% | 11% | 0% | 35% |
| | ASSET | 100% | 39% | 64% | 25% | 59% | 38% | 24% | 41% | 53% | 38% | 60% | 36% | 24% | 60% |
| | CD | 59% | 20% | 5% | 20% | 100% | 19% | 100% | 20% | 96% | 20% | 72% | 20% | 5% | 20% |
| | **GCBD** | **90%** | **2%** | **82%** | **13%** | **100%** | **3%** | **100%** | **8%** | **99%** | **1%** | **94%** | **5%** | **82%** | **13%** |
| Dataset | Attacks → | BadNets (3%) | | Input-aware (8%) | | FTrojan (0.8%) | | Naricissus (0.05%) | | SIBA (0.3%) | | Average | | Worst-Case | |
| | Detection↓ | TPR↑ | FPR↓ | TPR↑ | FPR↓ | TPR↑ | FPR↓ | TPR↑ | FPR↓ | TPR↑ | FPR↓ | TPR↑ | FPR↓ | TPR↑ | FPR↓ |
| CIFAR-100 | SS | 16% | 15% | 10% | 16% | 0% | 15% | 40% | 15% | 8% | 15% | 15% | 15% | 0% | 16% |
| | AC | 10% | 2% | 2% | 4% | 0% | 1% | 100% | 2% | 0% | 6% | 22% | 3% | 0% | 6% |
| | STRIP | 95% | 13% | 21% | 16% | 96% | 13% | 100% | 18% | 73% | 13% | 77% | 15% | 21% | 18% |
| | SentiNet | 0% | 0% | 1% | 0% | 100% | 100% | 0% | 0% | 0% | 0% | 20% | 20% | 0% | 100% |
| | ABL | 30% | 0% | 20% | 5% | 38% | 0% | 0% | 0% | 1% | 1% | 18% | 1% | 0% | 5% |
| | SCAN | 95% | 0% | 50% | 0% | 99% | 0% | 0% | 0% | 0% | 0% | 49% | 0% | 0% | 0% |
| | TeCo | 90% | 1% | 13% | 10% | 99% | 2% | 98% | 8% | 42% | 13% | 68% | 0% | 13% | 13% |
| | AGPD | 0% | 0% | 50% | 0% | 0% | 0% | 0% | 0% | 26% | 0% | 15% | 0% | 0% | 0% |
| | ASSET | 100% | 10% | 58% | 20% | 25% | 32% | 0% | 0% | 34% | 30% | 43% | 18% | 0% | 32% |
| | CD | 99% | 20% | 61% | 20% | 100% | 20% | 100% | 18% | 100% | 19% | 92% | 19% | 61% | 20% |
| | **GCBD** | **83%** | **8%** | **83%** | **2%** | **100%** | **7%** | **96%** | **14%** | **100%** | **1%** | **92%** | **6%** | **83%** | **14%** |

against Narcissus attacks. Furthermore, SIBA designs an effective trigger by formulating trigger generation as a bi-level optimization problem with sparsity and invisibility constraints. The trained triggers show the strong link with normal features, enabling SIBA to penetrate most defenses. In contrast, GCBD exhibits almost 100% TPR in detecting the triggers of SIBA and Narcissus.

## 4.2 PERFORMANCE OF OUR METHODS ON TEST-TIME DETECTION

Furthermore, we also present a test-time validation approach by testing the detector's classification accuracy (ACC) on the test set and its corresponding dataset poisoned by new attacks. ***Test-time detection means that GCBD has no access to the test images during training and can only perform independent analysis and classification based on the characteristics of backdoor features.*** According to Table 2, GCBD exhibits 90% ACC in most cases. Using ResNet34 as victim models,

Table 2: Performance of GCBD on test-time detection.

| Attack Models | Datasets → | CIFAR-10 | | | CIFAR-100 | | | Flag | |
|---|---|---|---|---|---|---|---|---|---|
| | Attacks ↓ | TPR↑ | FPR↓ | ACC↑ | TPR↑ | FPR↓ | ACC↑ | label-poisoning | characteristics |
| ResNet34 | Badnets (Gu et al. (2017)) | 0% | 1% | 49.97% | 99% | 4% | 97.65% | Poison-label | Same attack with different setting |
| | Input-aware (Nguyen & Tran (2020)) | 4% | 11% | 46.45% | 97% | 6% | 95.55% | Poison-label | Sample-specific dynamic trigger |
| | FTrojan (Chen et al. (2017)) | 99% | 17% | 91.16% | 99% | 15% | 91.89% | Poison-label | Frequency-stealthy trigger |
| | Narcissus (Zeng et al. (2023)) | 82% | 20% | 81.16% | 87% | 8% | 89.46% | Clean-label | Low poisoning rate |
| | SIBA (Gao et al. (2024)) | 45% | 1% | 72.06% | 93% | 16% | 88.23% | Clean-label | Low pixel variation |
| | Grond (Xu et al. (2025)) | 69% | 2% | 83.52% | 75% | 23% | 76.09% | Clean-label | Low model parpameter variation |
| | **Average** | 50% | 8% | 70.72% | 92% | 12% | 90% | - | - |
| ResNet18 | Badnets (Gu et al. (2017)) | 89% | 15% | 87.30% | 100% | 19% | 90.25% | Poison-label | Same attack with different setting |
| | Input-aware (Nguyen & Tran (2020)) | 97% | 16% | 90.31% | 85% | 10% | 87.71% | Poison-label | Sample-specific dynamic trigger |
| | FTrojan (Chen et al. (2017)) | 100% | 8% | 95.98% | 100% | 7% | 96.17% | Poison-label | Frequency-stealthy trigger |
| | Narcissus (Zeng et al. (2023)) | 94% | 5% | 94.70% | 98% | 5% | 96.59% | Clean-label | Low poisoning rate |
| | SIBA (Gao et al. (2024)) | 96% | 7% | 94.81% | 97% | 1% | 97.81% | Clean-label | Low pixel variation |
| | Grond (Xu et al. (2025)) | 100% | 2% | 99.00% | 98% | 1% | 98.47% | Clean-label | Low model parpameter variation |
| | **Average** | 96% | 9% | 93.68% | 96% | 7% | 94.50% | - | - |

GCBD achieves 71.72% ACC in CIFAR-10. After reducing the depth of the network to ResNet18, GCBD attains 94% ACC and 96% TPR. The false positive rate (FPR) is also maintained at approximately 8%. We hypothesize that deeper network models tend to average the probabilities in the final classification layer, thereby diminishing the disparity between backdoor features and benign features. Additionally, GCBD performs better on CIFAR-100, particularly in the ResNet34 scenario, as the rich category diversity may prevent models from averaging the probabilities in the classification layer. **In summary, GCBD can still detect unseen samples poisoned by new triggers in challenging test-time scenarios while maintaining satisfactory detection performance.**

The vast majority of backdoor attacks cannot be effectively and cost-efficiently transferred to the ImageNet. Therefore, we use the poison-label {Badnets, Blend} to train the GCBD at a subset of ImagnetNet, which is then used to detect the advanced Grond attack. GCBD achieves 100% TPR (0.83 F1) with 80% detection accuracy at test-time detection.

## 4.3 EFFECT OF TARGET LABELS & VALIDATION METHODS

In this section, we explore the effect of target classes and verification methods on the detection performance of GCBD through the FTrojan attack. Specifically, 2% samples of CIFAR-10 are randomly selected to be poisoned at the dirty-label settings with different target labels.

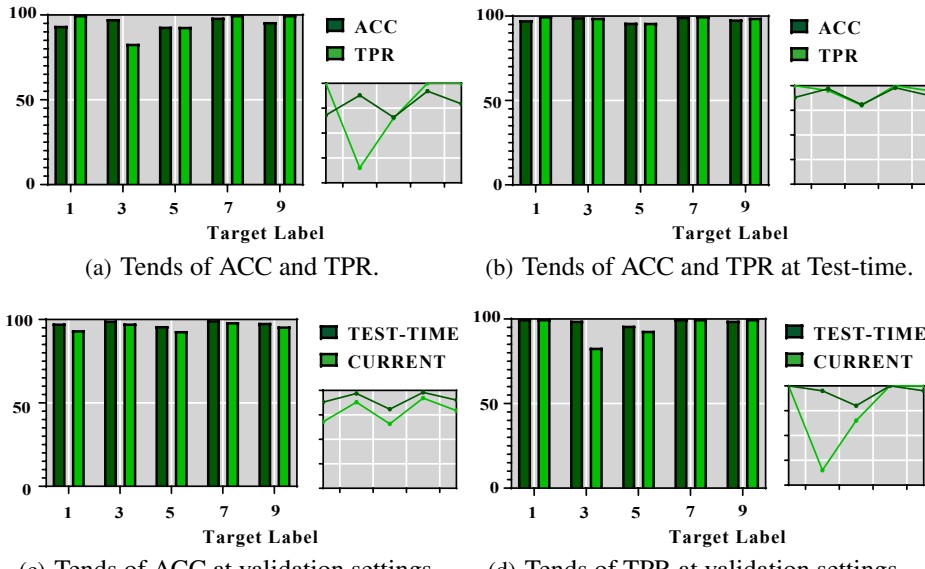

(a) Tends of ACC and TPR.

(b) Tends of ACC and TPR at Test-time.

(c) Tends of ACC at validation settings.

(d) Tends of TPR at validation settings.

Figure 3: Performance of GCBD on the FTrojan attack with different target labels.

Figure 3a illustrates the performance of GCBD in the mainstream validation method by detecting poisoned samples from the poisoned training set, which is named as **CURRENT** in Figures 3c and 3d. GCBD achieves approximately 90% classification accuracy (ACC) and true positive rate (TPR) for each target class. Notably, the fluctuation in TPR is slightly greater than that in ACC, which is attributable to the bias of TPR upon poisoned samples and the low proportion of poisoned samples in the train set. Specifically, TPR places greater emphasis on the accuracy of the detection method upon the trigger-implanted samples, while disregarding accuracy on normal images. **The extremely low poisoning ratio amplifies the randomness of the final detection performance**. Under the Test-time validation method, considering that each image in the test set has both a normal and a trigger-implanted version, there is no issue of a low proportion of poisoned samples, so that the influence of randomness is also circumvented. The stable TPR shown in Figure 3b validates the aforementioned hypothesis, exhibiting the superiority of the new testing method in terms of stability.

What is more, Figures 3c and 3d demonstrate the consistency between the two testing approaches. Both approaches can accurately reflect the performance of the detection methods. **It is noteworthy that test-time testing represents a more challenging detection scenario, as the detection methods cannot access specific information about the data to be detected during training**. Traditional detection methods are only applicable when all data information is available during testing, and thus cannot be used to evaluate real-time detection performance. In real-world backdoor attack scenarios, it is more common to lack access to all samples for validation.

**Performance of GCBD on multi-target attacks** Currently, there is little mature research on multi-target backdoor attacks published in top-tier conferences. This is because it breaks the traditional assumptions of backdoor attacks and, to a certain extent, overlaps with the objectives of adversary attacks, making it difficult to clearly define. Research on backdoor defense does not necessarily need to specifically cover this scenario.

Regarding whether there exists a feature space where various M-to-N backdoor attacks converge on a certain feature vector (i.e., whether there is a true commonality), we designed an experimentally enlightening experiment in the backdoor domain as follows: Firstly, in the training process of GCBD, we use 0 as the target category, which means we only need 1/10 (CIFAR10) and 1/100 (CIFAR100) of the data for GCBD training, representing a significant resource-saving advantage. Secondly, we perform a Badnets attack (poison-label) with a target category of 5 and a Narcissus attack (clean-label) with a target category of 3, and the choices of 5 and 3 are randomly generated. The setup implies that the trigger features are completely different (Badnets and Narcissus attacks) and the target categories are completely different (multi-label). What is more, the target category setting of GCBD training is different from all attacks. Within 10 epochs, GCBD achieves 99%TPR and 7% FPR (0.96 F1) at the detection of Badnets attack. Moreover, GCBD achieves 100%TPR

and 13% FPR (0.94 F1) at the detection of Narcissus attack. Therefore, even when GCBD is trained with completely different target labels and the dataset contains completely unrelated M-to-N attacks, GCBD still achieves satisfactory detection results within a few epochs.

## 4.4 Stability of GCBD & Deployment Analysis

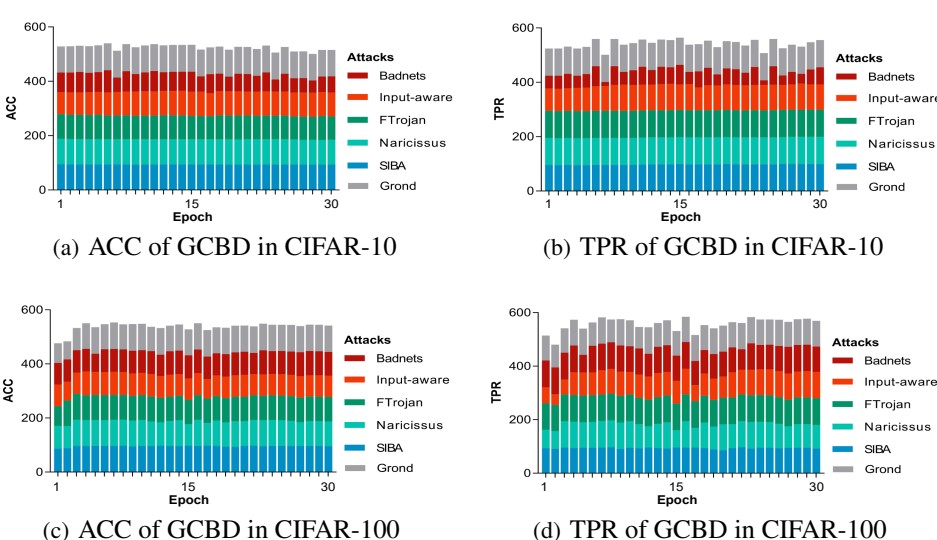

(a) ACC of GCBD in CIFAR-10

(b) TPR of GCBD in CIFAR-10

(c) ACC of GCBD in CIFAR-100

(d) TPR of GCBD in CIFAR-100

Figure 4: The test-time detection performance of GCBD during the training period.

We train an extremely minimalist LSTM network with 53K params within a few epochs to highlight the superiority of GCBD. To better demonstrate the overall performance of GCBD in detecting various attacks, we adopt the sum of Accuracy (ACC) as the y-axis variable. As illustrated in Figure 4, **GCBD achieves and subsequently sustains an average 93% ACC and 97% TPR after 4 epochs**. The remarkable stability of GCBD implies an exceptionally low computational cost. Specifically, the overhead for mainstream detection involves training the detector (*e.g.*, ResNet18 with 11.17M params) with 40 epochs on the entire dataset. In contrast, GCBD only needs a single class set (*e.g.*, $1/100$ of CIFAR-100) to learn Disturbance Immunity. With two traditional backdoor attacks labeling samples at the clean-label setting ($2 + 1$ **times** of the set), GCBD can be successfully trained within $4$ epochs ($1/10$ of the standard number of epochs). **Integrated with high-dimensionality sequences** ($m = 10, n = 5$)**, the cost of GCBD is much less than** $1/100 * 3 * 1/10 = 3/1000$ **of pixel-based (**$3 \times 32 \times 32$**) detectors**.

## 5 Conclusion

In this paper, we theoretically demonstrate that benign and backdoor features exhibit significant classification probability discrepancies across varying perturbations of clean image classes and intensities. Subsequently, conventional attack patterns are applied to train the proposed GCBD by transforming traditional unsupervised detection into a simple labeled binary classification task. What is more, the superiority of GCBD lies in its stepping out of the basic framework of feature separation and providing a detection method that directly migrates the commonalities of backdoor attacks. With few clean images ($\leq 10$), GCBD exhibits SOTA detection performance with satisfactory generalization on various SOTA attacks. Furthermore, we design a much more challenging test-time validation approach to exhibit the superiority of our work. To the best of our knowledge, **GCBD represents the first attempt to detect unseen samples poisoned by new triggers in challenging test-time scenarios while maintaining satisfactory detection performance**. Last but not least, current detection methods implicitly assume the existence of only one type of backdoor feature. Methods based on feature separation may overlook deeper-level triggers in scenarios where both shallow and deep triggers exist simultaneously.

## 6 REPRODUCIBILITY STATEMENT

During the publication phase, we will provide full access to all codes, logs, and result files to ensure transparency and reproducibility of our work.

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

## A  USAGE OF LLMs

Ultimately, we pledge that LLMs are applied to optimize descriptions in this paper. Furthermore, LLMs will not be employed for any purposes beyond those explicitly stated. All content generated by LLMs undergoes rigorous human verification and refinement to ensure factual accuracy.

## B  EFFECT OF CLEAN IMAGES

### B.1  SAMPLE SELECTION

Substantial evidence has demonstrated the varying levels of significance of sampleduring model training (Wu et al. (2025a)). Several studies (Hayase & Oh, 2022; Li et al., 2023; Li et al., 2024; Hung-Quang et al., 2024; Wang et al., 2025) have proposed sample selection approaches to enhance Attack Success Rate (ASR) by poisoning specific "hard" samples rather than random ones. These poisoned models tend to internalize implicit mappings between trigger features and target labels, thereby circumventing the original classification challenges associated with such samples.

Gao et al. (2023) reveals differential sample importance and selects "hard" samples via three metrics (*e.g.*, Forgetting Event, Loss Value, and Gradient Norm) to enhance the **P**oison-only **B**ackdoor **A**ttacks (PBAs). The poisoned models tend to learn the implicit projection between the trigger feature and the target label to evade the difficulty of the original classification upon such "hard" samples. For example, Forgetting Event identifies "hard" samples by analyzing misclassification transitions (i.e., shifts from correct to incorrect classification) during pre-training. Furthermore, category diversity is introduced to optimize the Forgetting Event metric with various intensities (*e.g.*, Res-linear), thereby enhancing improvements in ASR.

The aforementioned studies have thoroughly elucidated the significant impact of sample selection on backdoor attacks. Consequently, we investigate whether sample selection strategies closely associated with backdoor attacks exert an influence on the proposed detection approach. During the training process of GCBD, hard images of varying degrees are employed for dimensionality augmentation, and the experimental results can be seen in Figure 5. We compile the metrics of sample selection utilized in the relevant experiments as follows:

**Loss Value**  Given a surrogate model $f_\theta$ (trained on the poisoned training set), the loss value of the model on sample $(x_i, y_i)$ can be represented as $L(f_\theta(x_i), y_i)$. We choose samples with the highest value in each class $y_t \in \{y_0, y_1, \ldots, y_k\}$ as the source of disturbance:

$$x_s = arg \max_{x_s \subset D_t} L(f_\theta(x_s), y_t). \tag{17}$$

**Gradient Norm**  Given a surrogate model $f_\theta$ (trained on the poisoned training set), the $l_2-$ gradient norm of model on sample $(x_i, y_i)$ can be represented as $||\nabla_\theta L(f_\theta(x_i), y_i)||_2$. We choose samples with the highest value in each class $y_t \in \{y_0, y_1, \ldots, y_k\}$ as the source of disturbance:

$$x_s = arg \max_{x_s \subset D_t} ||\nabla_\theta L(f_\theta(x_s), y_t)||_2. \tag{18}$$

**Forgetting Event**  Given a sample $(x_i, y_t)$ in the target-label set $D_t$, Forgetting Event denotes the event when the sample is classified by the surrogate model from $y_t$ to $y_m(y_m \neq y_t)$, whose frequency can be represented as $Num_{forget}(x_i, y_m)$. We choose samples with the highest value in each class $y_t \in \{y_0, y_1, \ldots, y_k\}$ as the source of disturbance:

$$x_s = arg \max_{x_s \subset D_t} Num_{forget}(x_s, y_t). \tag{19}$$

**Category Diversity**  We use $\mu$ to represent the mean of Forgetting Event metric $\{Num_{forget}(x_i, y_m)(y_m \neq y_t)\}$. We choose samples with the highest value in each class $y_t \in \{y_0, y_1, \ldots, y_k\}$ as the source of disturbance:

$$x_s = arg \min_{x_s \subset D_t} \sum_{y_m \neq y_t} ||Num_{forget}(x_s, y_m) - \mu||_2. \tag{20}$$

"Res-X" represents a series of distinct negative functions $N_F$ to adjust weights of categories according to the Forgetting Event (frequency) at varied rates X ($O(\log(x)), O(x), O(x^2)), \ and \ O(e^x)$). Higher rates highlight the significance of Category Diversity in sample selection. For example, metric calculation with $N_F$ at $\log(x)$, dubbed Res-log, is depicted in Algorithm 2.

---

**Algorithm 2** Metric Calculation with Negative Function $N_F$ at $O(\log(x))$

---

**Input :** Train Dataset $D_{tr}$, Target Label $y_t$, Misclassification Events $Num_{forget}(x_i, y_m)$
**Output :** Calculated Metric of Samples
**for** $y_m \in labels$ **do**
$\quad Num[y_m] = \sum_{(x_i, y_t) \in D_{tr}} *Num_{forget}(x_i, y_m)$
**end for**
$Sum = \sum_{y_m \in labels} log(1 + Num[y_m])$
**for** $y_m \in Y$ **do**
$\quad Cls[y_m] = 1 - \frac{log(1 + Num[y_m])}{Sum}$
**end for**
**for** image $(x_i, y_t) \in D_{tr}$ **do**
$\quad Metric[x_i] = \sum_{y_m \in labels} Cls[y_m] * Num_{forget}(x_i, y_m)$
**end for**

---

## B.2 EXPERIMENTAL RESULTS

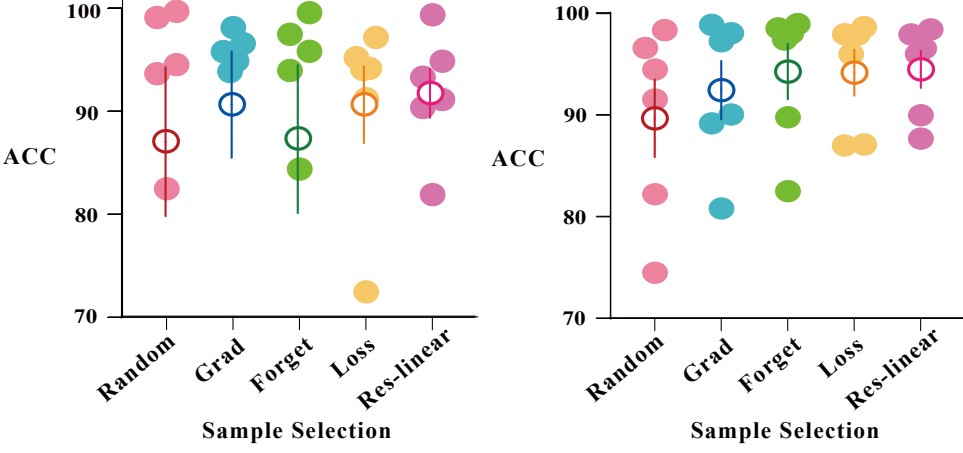

(a) Effect of sample selection in CIFAR-10    (b) Effect of sample selection in CIFAR-100

Figure 5: ACC of GCBD with different sample selections of clean images as the disturbance type.

As illustrated in Figure 5, utilizing 10 clean images selected via the Res-linear method as perturbation examples leads to a slight improvement in the detection accuracy (ACC) of the proposed method. Specifically, compared to the random selection approach, GCBD achieves an ACC increase of less than 5% when employing the Res-linear method. Moreover, the performance gap between the Res-linear method and other selection strategies is relatively narrow.

Notably, the Res-linear method significantly enhances the stability of the GCBD approach. On the CIFAR-10 dataset, when Random and Forget methods are used for image selection, the variance in detection ACC across different attacks is substantially higher than that observed with the Res-linear method. This same trend is also evident in the CIFAR-100 results. Overall, GCBD requires only 10 clean images as perturbations. Therefore, in practical applications, the stability of GCBD can be readily improved by manually selecting such challenging data samples. Even with random selection, GCBD maintains an at least 80% ACC.

As shown in Table 3, the Res-linear method achieves the best overall detection performance, with true positive rates (TPR) of 94.33% on CIFAR-10 and 96.33% on CIFAR-100. In the worst-case scenarios, the corresponding TPR values drop to 77.17% and 90.67%, respectively, resulting in

Table 3: Effect of sample selection methods upon the test-time detection performance of GCBD.

| Dataset | Attacks → | BadNets (3%) | | Input-aware (8%) | | FTrojan (2%) | | Narcissus (0.05%) | | SIBA (1%) | | Grond (1%) | | Average | |
|---|---|---|---|---|---|---|---|---|---|---|---|---|---|---|---|
| | Sample Selection↓ | TPR ↑ | FPR ↓ | TPR ↑ | FPR ↓ | TPR ↑ | FPR ↓ | TPR ↑ | FPR ↓ | TPR ↑ | FPR ↓ | TPR ↑ | FPR ↓ | TPR ↑ | FPR ↓ |
| CIFAR-10 | Random | 66% | 1% | 15% | 9% | 100% | 2% | 91% | 4% | 91% | 4% | 100% | 2% | 77.17% | 3.67% |
| | Grad | 98% | 5% | 41% | 11% | 100% | 12% | 96% | 4% | 96% | 4% | 100% | 4% | 88.50% | 6.67% |
| | Forget | 72% | 3% | 14% | 8% | 100% | 5% | 98% | 7% | 98% | 10% | 99% | 0% | 80.33% | 5.50% |
| | Loss | 90% | 8% | 64% | 19% | 100% | 6% | 97% | 9% | 95% | 5% | 100% | 12% | 91.00% | 9.83% |
| | Res-linear | 74% | 10% | 97% | 16% | 100% | 18% | 99% | 13% | 96% | 7% | 100% | 1% | 94.33% | 10.83% |
| | Res-square | 39% | 16% | 95% | 18% | 100% | 17% | 99% | 13% | 96% | 7% | 100% | 3% | 88.17% | 12.33% |
| | Res-log | 78% | 2% | 50% | 11% | 100% | 12% | 99% | 9% | 96% | 7% | 100% | 0% | 87.17% | 6.83% |
| | Average | 74% | 6% | 54% | 13% | 100% | 10% | 95% | 10% | 95% | 6% | 100% | 3% | 86.67% | 9.28% |
| Dataset | Attacks → | BadNets (3%) | | Input-aware (8%) | | FTrojan (0.8%) | | Narcissus (0.05%) | | SIBA (0.3%) | | Grond (0.6%) | | Average | |
| | Sample Selection↓ | TPR ↑ | FPR ↓ | TPR ↑ | FPR ↓ | TPR ↑ | FPR ↓ | TPR ↑ | FPR ↓ | TPR ↑ | FPR ↓ | TPR ↑ | FPR ↓ | TPR ↑ | FPR ↓ |
| CIFAR-100 | Random | 99% | 10% | 72% | 7% | 99% | 2% | 94% | 1% | 85% | 2% | 95% | 46% | 90.67% | 11.33% |
| | Grad | 100% | 19% | 69% | 7% | 99% | 1% | 99% | 3% | 98% | 3% | 81% | 2% | 91.00% | 6.33% |
| | Forget | 100% | 20% | 87% | 22% | 99% | 1% | 99% | 1% | 98% | 3% | 98% | 2% | 96.83% | 8.17% |
| | Loss | 98% | 24% | 77% | 3% | 98% | 1% | 99% | 3% | 97% | 1% | 92% | 0% | 93.50% | 5.33% |
| | Res-linear | 100% | 20% | 85% | 10% | 100% | 7% | 98% | 5% | 97% | 2% | 98% | 1% | 96.33% | 7.5% |
| | Res-square | 100% | 20% | 85% | 10% | 100% | 7% | 98% | 5% | 97% | 2% | 98% | 1% | 96.33% | 7.5% |
| | Res-log | 100% | 20% | 85% | 10% | 100% | 7% | 98% | 5% | 97% | 2% | 98% | 1% | 96.33% | 7.5% |
| | Average | 100% | 19% | 80% | 10% | 99% | 4% | 98% | 3% | 96% | 2% | 94% | 7% | 94.43% | 7.67% |

performance gaps of 17.16% and 5.66%. Notably, as the number of categories increases, the detection efficacy of GCBD improves from 86.67% to 94.43% as the rich category diversity may prevent models from averaging the probabilities in the classification layer. Therefore, GCBD can sufficiently catch the **Disturbance Immunity** of various triggers in backdoor attacks.

In addition, we placed particular emphasis on evaluating GCBD's performance under worst-case scenarios. When detecting non-shared dynamic triggers employed in Input-aware attacks, sample selection exerts a profound influence on GCBD's performance, with the best-case and worst-case true positive rates (TPRs) reaching 97% and 14%, respectively. Nevertheless, since GCBD requires only 10 clean images as perturbation sources, manually constructing hard-to-learn data samples is feasible. In practical settings, GCBD can thus circumvent worst-case scenarios using the aforementioned approach. Given that real-world applications typically involve datasets with a significantly larger number of classes, scenarios with merely 10 categories are rare, and the diversity inherent in large-scale datasets inherently reduces the likelihood of worst-case performance. **Collectively, the above factors contribute to the high deployability of GCBD in real-world applications.**

## C    EFFECT OF SOURCE ATTACKS' ASR

During the training process of GCBD, we employ the most conventional attacks {Badnets, Blend} to label the poisoned dataset. Subsequently, we train victim models based on the original poisoned dataset and elevate pixel values to a two-dimensional probability sequence matrix based on the Disturbance Immunity. In this chapter, we investigate the effect of the Attack Success Rate of the victim model, which serves as the dimensionality-elevating tool, on the detection performance of the ultimately trained GCBD. Meanwhile, we clarify the fact that the dataset used for training the victim model is already poisoned by the unseen attacks, and the training of victim models is entirely independent of the labeling process. This implies that we have the flexibility to adjust the poisoning rate to train victim models with any desired ASR. As illustrated in Figure 6, the shaded regions

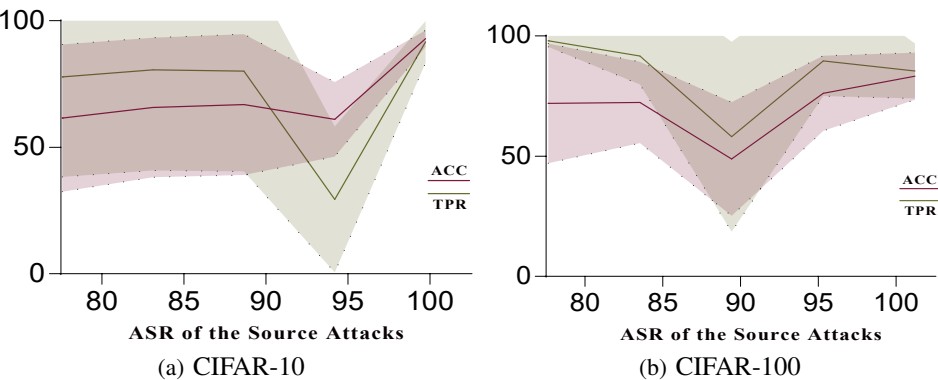

(a) CIFAR-10          (b) CIFAR-100

Figure 6: Effect of Source Attacks' ASR on GCBD.

indicate the fluctuation ranges, while the curves depict the variations in the average values of each metric. Initially, for GCBD, the average ACC and TPR metrics across various attacks exhibit a trend of first decreasing and then increasing with the rise in ASR. This suggests the interplay of at least two factors. Specifically, for CIFAR-10 and CIFAR-100, the metrics reach their lowest points when ASR is at 95% and 90%, respectively. Furthermore, the high ACC observed at low ASR levels fluctuates significantly depending on the type of attack being detected. Conversely, the high ACC at high ASR levels demonstrates remarkable stability.

Victim models that have not adequately learned backdoor features with low ASR levels tend to exhibit averaged probabilities on poisoned data. This blurs the distinct probabilistic characteristics of the triggers in Badnets and Blend attacks. Consequently, GCBD struggles to learn the commonalities (Disturbance Immunity) of the two attacks based on discernible differences, resulting in exceptional detection performance for triggers similar to Badnets or Blend, but near-total failure for others. In contrast, victim models that have thoroughly learned backdoor features display more distinctive probabilities on poisoned data, necessitating a deeper exploration of anti-interference properties to accomplish the training classification task. At this stage, the model captures truly generalized features, and its detection performance remains stable regardless of the attack type. The intermediate ASR scenario represents a dynamic confrontation between the aforementioned factors, where GCBD achieves its lowest ACC and TPR at the point of maximum conflict. Therefore, **we increase the poisoning rate to ensure that victim models can sufficiently learn traditional backdoor features for compelling GCBD to acquire Disturbance Immunity through the adversarial period**.

## D  EFFECT OF SOURCE ATTACKS' TYPE

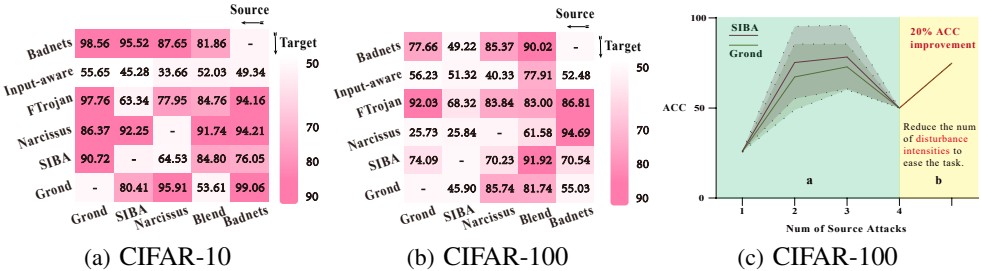

Figure 7: ACC of GCBD with different attacks as the single source to construct the train set.

As illustrated in Figures 7a & 7b, the detection performance of GCBD is highly contingent upon the trigger characteristics when employing only one type of attack as the source. Specifically, the difference between various cases for CIFAR-10 and CIFAR-100 can torch 65.4% (99.06% - 33.66%) and 68.96% (94.69% - 25.73%), respectively. We hypothesize that GCBD tends to overfit to the specific magnitude variations of the attack rather than its robustness when using a single attack. Thus, we progressively expanded the source based on the two worst-case scenarios in Figure 4c: detecting Narcissus using Grond and detecting Narcissus using SIBA. ACC initially rises and then declines with the increase in the number. We attribute this decline to the excessive complexity of the new dataset because GCBD achieves a 20% improvement in ACC when we mitigate the complexity by reducing the number of perturbation intensities, as depicted in part b of Figure 7c.

## E  EFFECT OF DISTURBANCE INTENSITIES

In this section, we investigate the effect of disturbance intensity sequences on the detection performance of GCBD. We employ the **Intensity Interval** to partition the range of $m$ (from $[0, 1]$), as defined in **Section** 3.2, into equal segments. For example, when the intensity interval is set to 11, the $m$ sequence becomes $\{0.0, 0.1, 0.2, \ldots, 1.0\}$. The results are illustrated in Figure 7, where each circle represents a detection of a specific attack.

As depicted in Figure 8, the mean values of ACC and TPR metrics for GCBD initially increase and then decrease as the Intensity Interval widens. Conversely, the variances of these metrics exhibit

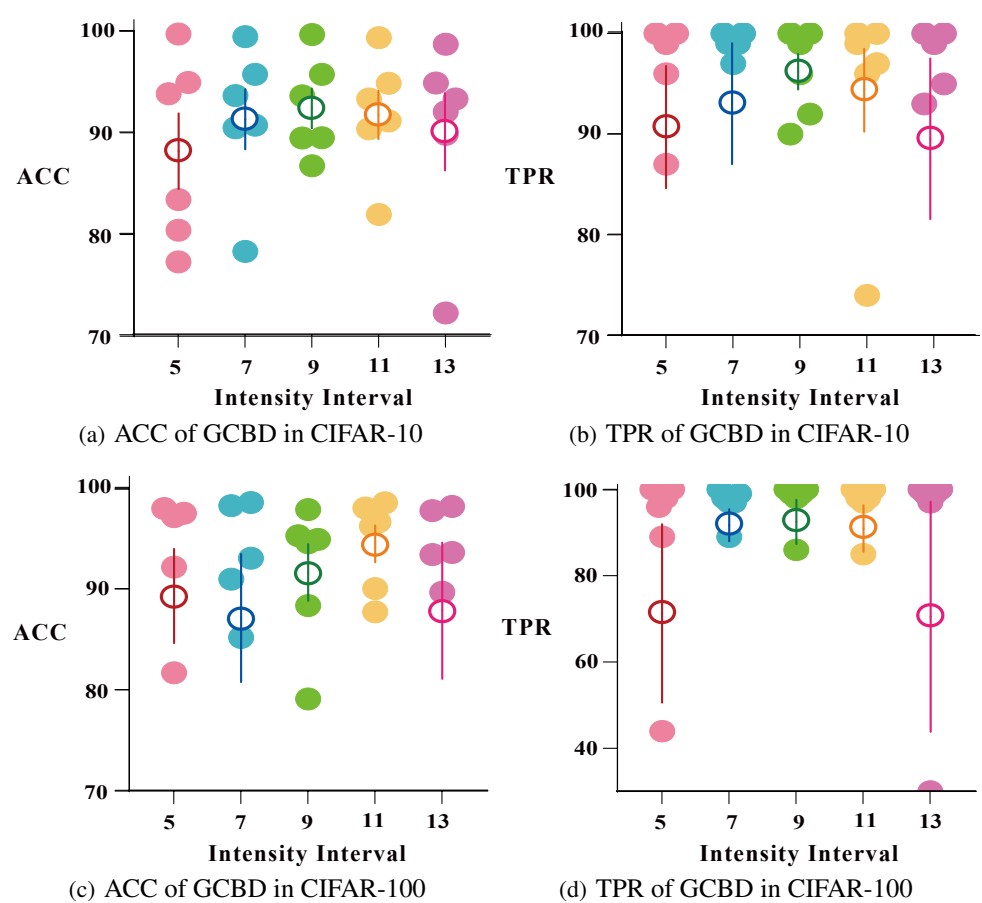

Figure 8: The test-time detection performance of GCBD during the training period.

(a) ACC of GCBD in CIFAR-10

(b) TPR of GCBD in CIFAR-10

(c) ACC of GCBD in CIFAR-100

(d) TPR of GCBD in CIFAR-100

an inverse trend, decreasing initially and then increasing. Consequently, GCBD achieves optimal detection performance when the Intensity Interval is set around 9. The above-observed phenomenon stems from the interplay of two key factors.

Firstly, when the Intensity Interval is relatively small, the overly broad intensity gaps result in significant information loss and an insufficient number of probabilities for processing. Therefore, GCBD falls into an overfitting state when learning disturbance immunity, thus producing poor performance. Conversely, when the Intensity Interval is excessively large, the overly narrow gaps introduce substantial redundant information and a high volume of probability data to be processed, increasing task complexity. Under such circumstances, the learning of the interference resistance by GCBD becomes underfitting, again leading to poor performance. **When the Intensity Interval is appropriately calibrated, GCBD can effectively learn the trigger's robustness against interference, achieving over 95% ACC and TPR**. In this paper, the default setting of Intensity Interval is 11.

## F    RELATED KNOWLEDGE IN BACKDOOR DETECTION

In this chapter, we provide supplementary explanations for the concepts and theoretical content presented in the main text to facilitate readers' comprehensive understanding of the division of labor during the model training and deployment phases.

### F.1    ATTACK KNOWLEDGE

In this attack setting, the adversary can operate on the original training dataset $D_{tr}$ and embed a predefined trigger into a small fraction of the training samples. The radio can be represented as the

poisoning rate. Furthermore, the attacks can be called clean-label attacks if the adversary refrains from altering the ground-truth labels of the original samples. However, the adversary lacks both the knowledge and the ability to modify other training components (*e.g.*, loss functions, model architectures, training schedules, or optimization algorithms). Consequently, attackers can only manipulate model weights through data poisoning. The latent association between the trigger and the target label is acquired solely during the training phase. During inference, we assume the adversary does not have access to the model's prediction vectors. Generally, poison-only clean-label attacks impose minimal requirements on the adversary's capabilities, making them applicable to a wide range of real-world scenarios.

## F.2 DETECTION KNOWLEDGE

The defender holds greater privileges than the attacker. Given a poisoned dataset, the defender can employ any measures to eliminate potentially embedded poisoned features. This implies that the defender has the autonomy to determine the architecture of a surrogate model and infer backdoor features based on the characteristics of the trained surrogate model, or alternatively, perform any operations directly on the poisoned dataset. Among these strategies, backdoor detection serves as an effective approach for backdoor defense, which can be conducted either during the pre-training phase or the inference phase by identifying samples containing triggers. However, the defender lacks information regarding the distribution of trigger-embedded samples, including the proportion of poisoned data, the characteristics of the triggers, and the inability to ascertain whether a sample is clean. Although clean data can be sourced from reliable external datasets, doing so incurs substantial additional costs. Consequently, for detection methods, a lower requirement for clean data translates to higher deployability. Meanwhile, most existing detection methods are confined to identifying trigger features within specific datasets. **To the best of our knowledge, GCBD represents the first cross-attack detector that can accurately detect entirely unknown triggers in new samples.**

## F.3 WORKFLOW OF THE DNNS

We detail the workflow of poison-only backdoor attacks and backdoor detection methods to formalize the generation of the final DNNs.

**Step 1: Sample Selection (by adversary).** Given a target label $y_t$, a subset $D_s$ is selected from target-label set $D_t = \{(x_i, y_i)|(x_i, y_i) \in D_{tr}, y_i = y_t\}$ to be poisoned. Therefore, the benign samples can be denoted as $D_b = D_{tr} \backslash D_s$. We represent the poisoning selection based on a binary vector $M = [M_1, M_2, \ldots, M_{|D_{tr}|}] \in \{0, 1\}^{|D|}$. Therefore, $M_i = 1$ when the smaple $x_i$ is selected to be poisoned and $M_i = 0$ means benign samples. The ratio of the poisoned samples $\alpha := \frac{|D_s|}{|D_{tr}|}$ is depicted as the poisoning rate. $\alpha$ can reflect the stealthiness of poison-only attacks. Backdoor attacks are supposed to maintain a high ASR with low $\alpha$ to evade possible machine and manual inspections. Furthermore, a low poisoning rate is equally essential for ensuring the normal functionality of DNNs, facilitating its deployment in real-world environments.

**Step 2: Trigger Design and Insertion (by adversary).** The requirement of stealthiness leads the adversary to carefully design a trigger pattern $w$ by tweaking the pixels of the images. Thus, the triggers can be applied to generate the poisoned images. The above period is depicted as $f_g : X \to X$. For example, $f_g(x) = (1 - m) * x + m * w$ is a common approach to implant the trigger $w$ where the mask $m \in [0, 1]^{C \times H \times W}$ represents the poison area of the images. $*$ represents the product in terms of elements. Given the target label $y_t$, the poisoned training dataset generated could be denoted as $D_p = \{(x_i, y_i)|_{if\ m_i=0},\ or\ (f_g(x_i), y_t)|_{if\ m_i=1}\}_{i=1}^{|D_{tr}|}$. Most attackers focus their efforts on designing triggers that can evade backdoor defenses and manual inspection. During the trigger design phase, attackers may also train surrogate models to validate and provide feedback for optimizing the triggers. However, they cannot make assumptions about the specific model architectures employed by defenders. Consequently, attackers must ensure that the backdoor attacks maintain effectiveness across a wide range of model architectures.

**Step 3: Backdoor Detection (by detectors).** Backdoor detection, serving as the initial step in the backdoor defense framework, holds significant research value. By eliminating high-risk data before they are actually utilized in model training, the implementation of backdoor attacks can be fundamentally thwarted. We represent the result of the detection based on a binary vector

$R = [R_1, R_2, \ldots, R_{|D_{tr}|}] \in \{0, 1\}^{|D|}$. Therefore, $R_i = 1$ when the smaple $x_i$ is classified as the poisoned sample and $R_i = 0$ means benign samples. Thus, the dataset $D_d$ used to be trained in Step 4 is constructed. Given that backdoor attacks can achieve a high ASR by poisoning only a minimal amount of data, the True Positive Rate (TPR) stands out as the most critical metric for detectors in this phase. Meanwhile, detectors must also aim for a low False Positive Rate (FPR) while maintaining high TPR, to prevent excessive exclusion of normal data from the training process.

**Step 4: Model Training (by defenders).** Once the final dataset $D_d$ is generated, users will train the DNN. The stealthiness and utility of backdoor attacks require imperceptible modifications of the data set, which require the poisoned model $\tilde{f}_\theta$ to maintain high accuracy in benign test data (high BA). Otherwise, users would not adopt the poisoned model, and no backdoor could be implanted. Meanwhile, during the training period, the defenders can employ various strategies to prevent the model from overfitting to the backdoor features. A wide array of defensive measures is available at this stage (*e.g.*, encompassing model compression, robustness training, and internal attack-defense drills). In the end, a final model will be trained, which has been subjected to backdoor attacks and is simultaneously being targeted for interception by defenders.

**Step 5: Inference stage of DNNs (by adversary and detectors).** The attackers expect to activate the injected backdoor using the trigger $w$ defined in Step 2. During the inference phase, adversaries attempt to utilize samples embedded with triggers to achieve illicit objectives or secure undue advantages. At this juncture, adversaries are solely privy to the final outcomes, lacking any access to the model's specific architecture, parameters, or the output information from each layer. In contrast, defenders can ascertain whether the data has been poisoned by inspecting model parameters and the output information from various layers. Upon identifying the data as high-risk, defenders can safeguard their interests by adopting measures such as service denial.

# G   EFFECT OF CATEGORY IN THE DESIGN OF GCBD

In the CIFAR-10 dataset, the mapping between y and the true labels is defined as {0:airplane, 1:automobile, 2:bird, 3:cat, 4:deer, 5:dog, 6:frog, 7:horse, 8:ship, 9:truck}. In Figure 6, we methodically organize the proportions of misclassified categories across diverse data categories, drawing particular attention to the most dominant category X based on the remarks "y=X". The accurate category corresponding to the pie chart, along with its identifying color relative to the other pie charts, is labeled above each visual representation. When samples from class A are frequently misclassified as class B, it indicates a notable resemblance between A and B.

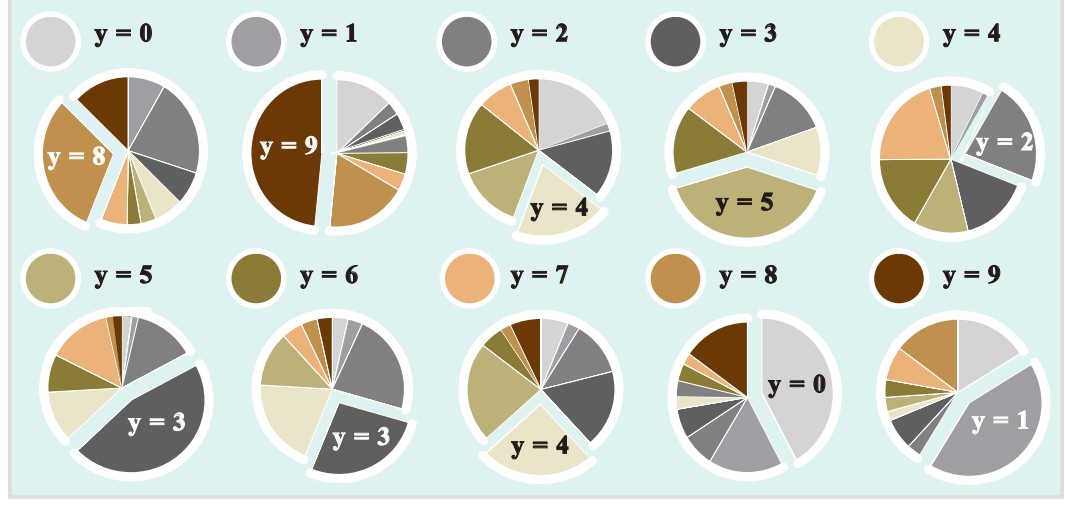

Figure 9: Category distinction within the CIFAR-10 dataset.

As depicted in Figure 9, there are significant disparities in the similarity levels across different categories. For example, the proportion of trucks (y=9) is considerably higher than that of birds (y=2).

Consequently, in the pie chart for automobiles (y=1), automobiles demonstrate a much stronger resemblance to trucks than to birds. Moreover, the similarity pattern exhibits symmetry. For the set y={0, 1, 2, 3, 4, 5, 8, 9}, the class with the highest proportion in its corresponding pie chart also predominates in the pie chart of the paired class. Although the set y={6, 7} deviates from this pattern, they still rank as the second-highest proportion in the corresponding pie charts for y={3, 4}. Therefore, to prevent the model from overfitting to shortcuts that distinguish backdoor features from normal ones based on the sequence of interfering categories, a process that diverts its focus away from learning adversarial robustness by exploiting category differences, **we frequently and randomly shuffle the order of interfering images during the training of the GCBD model**.

