# OpenReview forum: "General Cross-Attack Backdoor Detector Based on Disturbance Immunity of Triggers"
_ICLR.cc/2026/Conference — ICLR 2026 Conference Withdrawn Submission_

### Official Review · Reviewer_UV22 · 2025-10-28

**Soundness:** 2
**Presentation:** 1
**Contribution:** 2
**Rating:** 2
**Confidence:** 4

**Summary:**

This paper proposes GCBD (General Cross-attack Backdoor Detector), a novel backdoor detection method based on the concept of "Disturbance Immunity of triggers."  The key insight is that backdoor triggers must maintain their effectiveness across varying image perturbations, while benign features show different probability behaviors under such perturbations. GCBD requires only ≤10 clean images and demonstrates effectiveness across various attacks including Narcissus, SIBA, and Grond.

**Strengths:**

The paper focuses on the objective of backdoor attacks that the trigger can stably cause misclassification regardless of perturbations in the benign content. The proposed method focuses on the 'attacker's objective' rather than the 'attacker's artifacts', which is interesting.

**Weaknesses:**

- The paper should include experiments on high-resolution datasets, such as ImageNet, to demonstrate the method's scalability.

- The author only applied their method to models with the ResNet architecture. I am interested in knowing its generalizability on a wider variety of model structures.

- When using ResNet34 as the victim model on the CIFAR-10 dataset, GCBD performs extremely poorly.

- I believe Figure 4 has a significant error. Metrics like ACC should not be stacked, as this is misleading and the stacked values are meaningless. A line chart would be more intuitive for showing performance trends over epochs.

**Questions:**

- Did the authors only try LSTM as the sequence processing model? Was an ablation study conducted comparing LSTM with other models capable of processing sequential data (e.g., GRU, RNN, or Transformer)?

- If a trigger does not possess "disturbance immunity," would GCBD classify it as clean?

---

> ### Author Response · Authors · 2025-12-01
>
> W1
> >- The same as Q3 in "Rebuttal to Reviewer nwX5 (2)"
>
> {W2,W3,Q1}
> >- Our advantage lies in training with an extremely minimalist LSTM network within a very small number of epochs. Moreover, as the defending side, we have control over the structure of the detection model, whereas the attacking side needs to focus on compatibility with diverse victim models. If every proposed defense method requires extensive adaptation to different network structures, it would not only be impractical and meaningless but also trap research in a lengthy iterative process.
>
> W4
> >- To better demonstrate the overall performance of GCBD in detecting various attacks, we adopt the sum of Accuracy (ACC) as the y-axis variable. This is not an erroneous or meaningless design.
>
> W5
> >- Firstly, the questions raised by the reviewer are based on assumptions, mere negation, and skepticism. If you believe that certain types of attack methods are not covered, you should directly provide specific examples rather than resorting to purely subjective ones.
>
> >- The reviewer's misunderstandings regarding the observational conclusions stem from the following aspects, which we will explain one by one:
>
> >>-  Direct Assumption Issue
> The backdoor attack itself is a task with the objective of making the victim model classify images with triggers into a target category. Any research on backdoor attacks is designed based on this fundamental objective. Therefore, the two observational conclusions merely refine the description by categorizing images based on whether they belong to the same category or not, rather than making unfounded direct assumptions.
>
> >>- Misunderstanding Regarding the Specific Form of Triggers
> The reviewer uses the pixel-level diversity of dynamic triggers to deny the assumption of a single trigger. The triggers we refer to are conceptual descriptions that modify image features into attacker-specific features in a high-dimensional feature space, rather than triggers that are exactly the same at the pixel level. Our main experiments still cover pixel-level dynamic trigger attacks (Input-aware). Compared to the other 10 detection methods, GCBD demonstrates the best performance in detecting dynamic triggers, proving that our design is not limited to pixel-level consistency of triggers.
>
> >>- Multi-target Backdoor
> Currently, there is little mature research on multi-target backdoor attacks published in top-tier conferences. This is because it breaks the traditional assumptions of backdoor attacks and, to a certain extent, overlaps with the objectives of adversary attacks, making it difficult to clearly define. Research on backdoor defense does not necessarily need to specifically cover this scenario. It might be our general description that caused the reviewer's concern. In A4, we specifically designed a simulated multi-target backdoor attack to address the reviewer's doubts.
>
> >>- Regarding whether there exists a feature space where various M-to-N backdoor attacks converge on a certain feature vector (i.e., whether there is a true commonality), we designed an experimentally enlightening experiment in the backdoor domain as follows:
>
> >>>- Firstly, in the training process of GCBD, we use 0 as the target category, which means we only need 1/10 (CIFAR10) and 1/100 (CIFAR100) of the data for GCBD training, representing a significant resource-saving advantage.
> Secondly, we perform a Badnets attack (poison-label) with a target category of 5 and a Narcissus attack (clean-label) with a target category of 3 on the data, where the choices of 5 and 3 are randomly generated.
> The setup in 2 implies that the trigger features are completely different (Badnets and Narcissus attacks) and the target categories are completely different (multi-label). The target category setting of GCBD is different from that of the attacks (meaning GCBD does not need to know the real target categories).
> The experimental results are as follows:
> | attacks   | epoch | TPR  | FPR  | F1   | epoch | TPR  | FPR  | F1   | epoch | TPR  | FPR  | F1   |
> |-----------|-------|------|------|------|-------|------|------|------|-------|------|------|------|
> | Badnets   | 1     | 1.00 | 0.14 | 0.93 | 6     | 1.00 | 0.11 | 0.95 | 11    | 0.99 | 0.07 | 0.96 |
> | Narcissus | 1     | 1.00 | 0.27 | 0.88 | 6     | 1.00 | 0.20 | 0.91 | 11    | 1.00 | 0.13 | 0.94 |
>
> >>>- Conclusion Analysis:
> Even when GCBD is trained with completely different target labels and the dataset contains completely unrelated M-to-N attacks, GCBD still achieves satisfactory detection results within a few epochs.
>
> Conclusion
> >- This reviewer has focused on trivial details and an obsessive concern for the completeness of ablation experiments, while selectively ignoring the core innovative points, and then gave a score of 2.

---

### Official Review · Reviewer_PvKf · 2025-10-30

**Soundness:** 2
**Presentation:** 2
**Contribution:** 2
**Rating:** 2
**Confidence:** 4

**Summary:**

This paper introduces GCBD (General Cross-Attack Backdoor Detector), a detection framework aimed at identifying poisoned samples across various backdoor attacks by exploiting what the authors term the Disturbance Immunity property of triggers. The key intuition is that backdoor features exhibit consistent classification behavior under image perturbations, while benign features vary more. The authors claim to theoretically demonstrate this property and then reformulate detection as a binary classification problem rather than unsupervised clustering. GCBD is trained on a few clean samples and has been demonstrated across multiple datasets and attacks.

**Strengths:**

The paper tackles an important problem, which is detecting poisoned data while reducing dependency on large clean datasets.

The proposed defense is evaluated across both static and dynamic triggers.

**Weaknesses:**

**1. Conceptual and methodological clarity**

The overall pipeline is unclear and conceptually confusing. The paper lacks a coherent and complete description of how GCBD operates end-to-end. Starting with Algorithm 1, the pseudocode does not clearly specify the algorithm's final output, how the so-called "sequence dataset" (D) is constructed and used during training, or how the resulting model generates the final detection scores. It is also unclear what model is being trained in Algorithm 1. Is it the LSTM-based classifier mentioned elsewhere in the text? Furthermore, the paper does not explain how the extracted "scores" or "sequences" from the victim model are transformed into binary detection decisions.
In addition, the defense pipeline itself, including whether it is applied once or iteratively during the sanitization process, is never explicitly defined. Even Figures 1 and 2, which are meant to illustrate the overall framework, are visually dense and fail to convey a clear flow of information. In particular, it remains unclear how the disturbance generation, labeling, and classification stages interact within the proposed mechanism.

**2. Missing source code and reproducibility**

To further complicate understanding of the proposed framework, the authors do not share their code, which could clarify the pipeline and the rationale for many key experimental details (e.g., hyperparameters, data generation specifics, trigger parameters) that are missing.

**3. Theoretical claims without proofs**
Although the paper repeatedly claims to “theoretically demonstrate” the Disturbance Immunity property, the provided “Observations 1 and 2” are actually assumptions, not proofs. In particular, no formal derivation or theorem is provided beyond high-level equations that restate assumptions. Hence, the theoretical contribution is misrepresented. These are empirical intuitions, not proven results. The theoretical contribution claim is thus false and overstated.

**4. Weak experimental gain**

The paper shows limited or marginal improvements over strong baselines such as TeCo and AC when considering dynamic or unseen triggers (e.g., Input-Aware attack). The “cross-attack” claim is therefore overstated.
5. Experimental inconsistencies and weak justification
The poisoning rates used across attacks are inconsistent (ranging from 0.05% to 8%) without justification or tuning rationale. It is unclear whether these choices are fair or matched to prior baselines.
The proposed “test-time detection” setup (Section 4.2) is poorly explained: when does the defender have access to the test images? How does this differ from the prior evaluation in Section 4.1? The purpose of this stage is ambiguous, and it remains unclear whether it represents a realistic deployment scenario.

**5. Questionable novelty**

The claimed “new perspective” that backdoor effects rely on trigger presence rather than image content is not new. This is already well established in the backdoor literature (e.g., BadNets, SentiNet). Thus, the conceptual novelty of “Disturbance Immunity” is limited; it repackages a known observation with new terminology but without new theoretical or empirical depth.

**Minor**:  Figure 3 is missing the x and y-axis description.

**Actionable Points**
Here is a list of actionable points for the authors to improve the paper from my perspective:

- Provide a precise, formal, and algorithmic description of the GCBD framework.
- Reframe Observations 1 and 2 as hypotheses or assumptions and remove any claim of theoretical proof unless formal derivations are added.
- Ensure consistent experimental settings across all baselines, and include more detailed hyperparameter choices.
- Explain when test-time detection is applicable, its difference from training-phase detection, and whether it represents a realistic use case.

**Questions:**

How is the defense classifier trained?

Is the proposed defense applied once to the dataset for sanitization, or iteratively?

How were poisoning percentages chosen across attacks?

How does GCBD handle cases where new triggers differ structurally from those used in training? The Input-Aware results suggest no competitive advantage over the other approaches at the state of the art.

Since Observations 1–2 are assumptions, not formal proofs, can the authors provide any empirical ablation to support these claims?

---

> ### Author Response · Authors · 2025-12-01
>
> {W1, Q1, Q2, W2}
> >- We will provide the code after the article is published, and the complete implementation details will be fully disclosed. The paper should focus on explaining the core components rather than presenting the entire engineering process.
>
> >- The commitment to code reproducibility and open-sourcing is already included in ICLR's REPRODUCIBILITY STATEMENT, so it should not be listed as a weakness in the reviews.
>
> {W3, Q5}
> We have revised the relevant statements to avoid "overstating the case."
>
> >- The reviewer's misunderstandings regarding the observational conclusions stem from the following aspects, which we will explain one by one:
>
> >>- Direct Assumption Issue
> The backdoor attack itself is a task aimed at leading the victim model to classify images with triggers into a target category. Any research on backdoor attacks is designed based on this fundamental objective. Therefore, the two observational conclusions merely refine the description by categorizing images based on whether they belong to the same category or not, rather than making unfounded direct assumptions.
>
> >>- Misunderstanding Regarding the Specific Form of Triggers
> The reviewer uses the pixel-level diversity of dynamic triggers to deny the assumption of a single trigger. The triggers we refer to are conceptual descriptions that modify image features into attacker-specific features in a high-dimensional feature space, rather than triggers that are exactly the same at the pixel level. Our main experiments still cover pixel-level dynamic trigger attacks (Input-aware). Compared to the other 10 detection methods, GCBD demonstrates the best performance in detecting dynamic triggers, proving that our design is not limited to pixel-level consistency of triggers.
>
> {W4, Q4}
>
> >- It is a bias for the reviewer to focus solely on a specific result of the detection method under a particular attack while ignoring the average and worst-case scenarios. Teco performs extremely poorly in detecting Narcissus attacks, with only 46% TPR and 20% FPR, yet this is not mentioned at all. Furthermore, in our experiments based on BackdoorBench, the AC method performs poorly, but the reviewer considers it a strong baseline, which reflects that the reviewer has not carefully examined the experimental results, and yet gives a conclusion that the improvement is insignificant.
>
> W5
>
> >-  Anti-interference capability is a transformation of the description of the objectives of backdoor attack tasks. However, currently, no method provides detection work based on direct commonality transfer designed on this basis, nor does it break the feature separation idea with a labeled-conversion detection framework. It is unconvincing for the reviewer to completely deny the innovation due to anti-interference capability.
>
> Q3
>
> >- The strength of backdoor features varies among different backdoor attacks. We use a 95% Attack Success Rate (ASR) as the criterion for an effective backdoor attack. Different poisoning rates represent the coarse-grained minimum poisoning rates corresponding to different attacks that meet this criterion.

---

### Official Review · Reviewer_fyLB · 2025-10-31

**Soundness:** 1
**Presentation:** 1
**Contribution:** 1
**Rating:** 0
**Confidence:** 4

**Summary:**

This paper proposed a backdoor detector based on the disturbance immunity of triggers. Specifically, they give two assumptions: 1) disturbance immunity across classes: any original sample from different classes, the poisoned version’s feature close to a fixed vector; and 2) disturbance immunity across intensities: for any sample from a same class, the poisoned samples’ feature can be divided to a fixed vector and a distortion with tiny norm than the fixed one.

**Strengths:**

Strength:

1.	Systematical comparison: The authors evaluate their method with many existing defenses, by checking the TPR, FPR of the poison detection.

2.	Theoretical analysis: They formally argue the so-called disturbance immunity in a formal way.

**Weaknesses:**

Weakness:

1.	It is a little bit hard to follow this paper and the writing should be improved significantly.

2.	In Section 3 the proposed work, the authors first talk about the basic knowledge on the trigger injection, which is not originally proposed by the authors. I suggest moving this part to the preliminary.

3.	For the disturbance immunity across intensities, the picture 2 show the intensity is the blending ratio of the trigger signal, but in observation 2 I don’t find any intensity definition. According to what the authors said, the observation 2 is very similar as observation 1 except limiting the x from one class.

4.	In Equ 10, the authors define the perturb_x:=(1-m)x and then in Equ 11 and 12, they said the (1-m) will not affect the prediction. It is possible right when x\approx 0 but once when it close to 1, this claim will definitely wrong.

5.	The authors claim the LSTM is trained as poison detector, but in section 3, I don’t see any description on training. Moreover, the definition on the ‘high dimensional sequences’ are missing.

6.	Lack of the extension: the LSTM analyzes the features to determine whether a sample is poisoned or not. This means the trained LSTM only work for a fixed feature space. In real cases, they feature space are various with the change of the architecture.

7.	For the Naricissus attack, people have proved it more like an adversarial example attack instead of backdoor.

**Questions:**

See the previous weakness

---

> ### Author Response · Authors · 2025-12-01
> **Rebuttal  to Reviewer fyLB**
>
> I believe it is unprofessional for the reviewer to give a score of 0 on the grounds of being unable to understand the article. Their misunderstanding about triggers is confined to the notion of pixel-level exact consistency, and we have already clarified the issues in our "Rebuttal to Reviewer nwX5".
>
> Our advantage lies in training with an extremely minimalist LSTM network within a very small number of epochs. Moreover, as the defending side, we have control over the structure of the detection model, whereas the attacking side needs to focus on compatibility with diverse victim models. If every proposed defense method requires extensive adaptation to different network structures, it would not only be impractical and meaningless but also trap research in a lengthy iterative process.

---

### Official Review · Reviewer_nwX5 · 2025-10-31

**Soundness:** 2
**Presentation:** 2
**Contribution:** 2
**Rating:** 2
**Confidence:** 4

**Summary:**

The paper introduced a General Cross-attack Backdoor Detector (GCBD), which provided a simple and efficient new direction for cross-attack backdoor detection through the unified perspective of disturbance immunity. The paper not only verifies on the traditional training set detection, but also designs a more practical test-time detection scenario. The experiments prove that although the detector is only trained on the most basic badNets/blend attacks, it can successfully detect multiple unseen attacks.

**Strengths:**

1. The paper transform unsupervised clustering detection into a supervised binary classification problem. Synthesize labeled samples using two conventional backdoors (BadNets, Blend) to train a lightweight LSTM detector to distinguish stable under perturbation. In this way, defenders do not need any attack prior or large-scale clean data, and can train a universal detector with only a small number of samples.
2. The accuracy and TPR of GCBD demonstrated high stability during both the training and testing phases

**Weaknesses:**

1. There is lack of experiment or theory to demenstarte the proposed two observations. Observation 1 is direct assumption, not derivation.  And this does not always hold true in fact. For example: 1) Input-aware attacks generate triggers of different styles (dynamic triggers) for different inputs, triggers may not be the same in the feature space 𝑣∗. 2)Multi-objective backdoors (jumping from different source classes to different target classes) clearly do not satisfy the same 𝑣∗.
2. The paper claim that  they theoretically demonstrate that benign and backdoor features exhibits ignificant classification probability discrepancies across varying perturbations of clean image classes and intensities, but there is no lower bound of any margin is used to quantify this separability.
3. there is a lack of discussion about the difference between GCBD and existing defense methods, such as STRIP.

**Questions:**

1. The paper only discussed all-to-one backdoor attacks. Can GCBD still work in all-to-all backdoor attacks? The proposed perturbation immunity hypothesis suggests that if each trigger corresponds to a different target label, this unified target class adsorption effect will be broken, so that GCBD may no longer be applicable?
2. How about the adaptive attacks? The paper is presented as a defense, but there is no discussion or analysis about adaptive attacks.
3. The method is only evaluated on low-resolution datasets and small models. This may limit the method's interest to a broader audience. How about the performance of GCBD on ImageNet and ViT?

---

> ### Author Response · Authors · 2025-12-01
> **Rebuttal to Reviewer nwX5**
>
> {W1}. The reviewer's misunderstandings regarding the observational conclusions stem from the following aspects, which we will explain one by one:
>
> >- Direct Assumption Issue
>
> The backdoor attack itself is a task to lead the victim model to classify images with triggers as the target label. Any research on backdoor attacks is designed based on this fundamental objective. Therefore, the two observational conclusions merely refine the description by categorizing images based on whether they belong to the same category or not, rather than making unfounded direct assumptions.
>
> >- Misunderstanding Regarding the Specific Form of Triggers
>
> The reviewer uses the pixel-level diversity of dynamic triggers to deny the assumption of a single trigger. The triggers we refer to are conceptual descriptions that modify image features into attacker-specific features in a high-dimensional feature space, rather than triggers that are exactly the same at the pixel level. Our main experiments still cover pixel-level dynamic trigger attacks (Input-aware). Compared to the other 10 detection methods, GCBD demonstrates the best performance in detecting dynamic triggers, proving that our design is not limited to pixel-level consistency of triggers.
>
> >- Multi-target Backdoor
>
> Currently, there is little mature research on multi-target backdoor attacks published in top-tier conferences. This is because it breaks the traditional assumptions of backdoor attacks and, to a certain extent, overlaps with the objectives of adversary attacks, making it difficult to clearly define. Research on backdoor defense does not necessarily need to specifically cover this scenario. It might be our general description that caused the reviewer's concern. We specifically designed a simulated multi-target backdoor attack to address the reviewer's doubts.
>
> >- Regarding whether there exists a feature space where various M-to-N backdoor attacks converge on a certain feature vector (i.e., whether there is a true commonality), we designed an experimentally enlightening experiment in the backdoor domain as follows:
>
> >> Firstly, in the training process of GCBD, we use 0 as the target category, which means we only need 1/10 (CIFAR10) and 1/100 (CIFAR100) of the data for GCBD training, representing a significant resource-saving advantage.
> Secondly, we perform a Badnets attack (poison-label) with a target category of 5 and a Narcissus attack (clean-label) with a target category of 3, and the choices of 5 and 3 are randomly generated.
>
> >> The setup implies that the trigger features are completely different (Badnets and Narcissus attacks) and the target categories are completely different (multi-label). The target category setting of GCBD training is different from all attacks.
>
> The experimental results are as follows:
> | attacks   | epoch | TPR  | FPR  | F1   | epoch | TPR  | FPR  | F1   | epoch | TPR  | FPR  | F1   |
> |-----------|-------|------|------|------|-------|------|------|------|-------|------|------|------|
> | Badnets   | 1     | 1.00 | 0.14 | 0.93 | 6     | 1.00 | 0.11 | 0.95 | 11    | 0.99 | 0.07 | 0.96 |
> | Narcissus | 1     | 1.00 | 0.27 | 0.88 | 6     | 1.00 | 0.20 | 0.91 | 11    | 1.00 | 0.13 | 0.94 |
> >>- Conclusion Analysis:
> Even when GCBD is trained with completely different target labels and the dataset contains completely unrelated M-to-N attacks, GCBD still achieves satisfactory detection results within a few epochs.
>
> {W2.}
> >- The reviewer's requirement for a theoretical lower bound on the interval is unreasonable. There has been no major breakthrough in the theoretical analysis of lower bounds for deep learning networks, and requiring such analysis in the field of backdoor attacks is even more difficult to understand. Most backdoor attack research does not require providing such a theoretical analysis.
>
> {W3}
>
> >- It is unnecessary to compare each of the 10 detection methods used for comparison one by one. We focus on explaining the core principles of GCBD, and other works should be studied in their original papers. However, we provide the differences between GCBD and general defense methods as follows:
>
> >>- The detection methods (e.g., GCBD) focus on the detection of poisoned samples. Other defense methods may focus on detecting part of the poisoned samples and then using reverse engineering to erase backdoor features, requiring lower accuracy in detection. Detection methods aim to eliminate backdoor features from the source and can also be applied in the inference phase of the models.
>
> >>- The superiority of GCBD lies in its stepping out of the basic framework of feature separation and providing a detection method that directly migrates the commonalities of backdoor attacks.
>
> >>- Current detection methods implicitly assume the existence of only one type of backdoor feature. Methods based on feature separation may overlook deeper-level triggers in scenarios where both shallow and deep triggers exist simultaneously.

---

> ### Author Response · Authors · 2025-12-01
> **Rebuttal to Reviewer nwX5 （2）**
>
> Q1.
> See W1.
>
> Q2.
> We solely focus on the field of backdoor attacks and, at the same time, provide a demonstration of detecting backdoor attacks with dynamic triggers (input-aware).
>
> Q3.
> Research on backdoor attacks on visual Transformer (ViT) models represents a specialized research direction and embodies the value of a single innovative point among the latest achievements in the recent expansion of backdoor attacks to large datasets, multiple domains, and ViT models. However, mainstream backdoor attack research does not need to cover every aspect.
>
> The vast majority of backdoor attacks we tested cannot be effectively and cost-efficiently transferred to the ImageNet or ViT scenarios (either because the authors have not open-sourced their code or the trigger strength does not support such transfer). Therefore, it is unnecessary to conduct multi-type backdoor attack detection on large datasets.
>
> ***However, we find that the poison-label Badnets(2017), the poison-label Blend (2017) , Grond (2025) attacks are effective in the subset of the Imagenet Dataset. Therefore, we train the GCBD based on \{Badnets, Blend\} and detect the advanced Grond attack. We hope the results can relieve the concerns. We achieve 100% TPR (0.83 F1) with 80% accuracy at test-time detection.***

---

### Author Response · Authors · 2025-12-01

We appreciate ICLR's mechanism that makes all reviews publicly available. We have carefully responded to all significant questions raised. During this review process, we received review comments with scores (2220), but we are no longer preoccupied with whether our paper will be accepted. We hope that researchers in the field of backdoor attacks can draw inspiration from our work and sincerely seek opinions and suggestions to advance together.

---

### Note · Authors · 2025-12-14

I have read and agree with the venue's withdrawal policy on behalf of myself and my co-authors.